Genome-wide identification, characterization and expression analysis of HAK genes and decoding their role in responding to potassium deficiency and abiotic stress in Medicago truncatula

Zhao Yanxue 1
Wang Lei 1
Zhao Pengcheng 2
Liu Zhongjie 3
Guo Siyi 1
Li Yang 1 liyang0378@henu.edu.cn
Liu Hao 1 10140026@vip.henu.edu.cn
1 State Key Laboratory of Crop Stress Adaptation and Improvement, School of Life Sciences, Henan University , Kaifeng , China
2 College of Grassland Science, Nanjing Agricultural University , Nanjing , China
3 Key Laboratory of Genetics and Fruit Development, College of Horticulture, Nanjing Agricultural University , Nanjing , China
Orlov Yuriy
Electronic publication date: 2022 Sep 22
Publication date: 2022
Volume: 10
Electronic Location ID: e14034
Received 2022 Mar 15; Accepted 2022 Aug 18
Copyright: © 2022 Zhao et al.
Copyright year: 2022
Copyright holder: Zhao et al.
License: This is an open access article distributed under the terms of the Creative Commons Attribution License, which permits unrestricted use, distribution, reproduction and adaptation in any medium and for any purpose provided that it is properly attributed. For attribution, the original author(s), title, publication source (PeerJ) and either DOI or URL of the article must be cited.
License URL: https://creativecommons.org/licenses/by/4.0/

Keywords: Genome-wide analysis, Medicago truncatula, HAKs, Expression pattern

Funding: National Natural Science Foundation of China 31970808 Henan Science and Technology Development Plan Project 212300410023 Program for Innovative Research Team (in Science and Technology) in University of Henan Province 21IRTSTHN019 University Key Scientific Research Project Plan of Henan Province 22A180001 This work was supported by the National Natural Science Foundation of China (31970808), the Henan Science and Technology Development Plan Project (212300410023), the Program for Innovative Research Team (in Science and Technology) in University of Henan Province (21IRTSTHN019), and the University Key Scientific Research Project Plan of Henan Province (22A180001). The funders had no role in study design, data collection and analysis, decision to publish, or preparation of the manuscript.

==============================
Background

The HAK family is the largest potassium (K+) transporter family, vital in K+ uptake, plant growth, and both plant biotic and abiotic stress responses. Although HAK family members have been characterized and functionally investigated in many species, these genes are still not studied in detail in Medicago truncatula, a good model system for studying legume genetics.

Methods

In this study, we screened the M. truncatula HAK family members (MtHAKs). Furthermore, we also conducted the identification, phylogenetic analysis, and prediction of conserved motifs of MtHAKs. Moreover, we studied the expression levels of MtHAKs under K+ deficiency, drought, and salt stresses using quantitative real-time PCR (qRT-PCR).

Results

We identified 20 MtHAK family members and classified them into three clusters based on phylogenetic relationships. Conserved motif analyses showed that all MtHAK proteins besides MtHAK10 contained the highly conserved K+ transport domain (GVVYGDLGTSPLY). qRT-PCR analysis showed that several MtHAK genes in roots were induced by abiotic stress. In particular, MtHAK15, MtHAK17, and MtHAK18 were strongly up-regulated in the M. truncatula roots under K+ deficiency, drought, and salt stress conditions, thereby implying that these genes are good candidates for high-affinity K+ uptake and therefore have essential roles in drought and salt tolerance.

Discussions

Our results not only provided the first genetic description and evolutionary relationships of the K+ transporter family in M. truncatula, but also the potential information responding to K+ deficiency and abiotic stresses, thereby laying the foundation for molecular breeding of stress-resistant legume crops in the future.

Introduction

Potassium (K+) is an essential macronutrient for various plant physiological functions, like ion homeostasis and the transport of nitrate and sugars (Li et al., 2018). Due to the limited K+ resource, plants have evolved a series of K+ transport systems to mediate its uptake and transport (Amrutha et al., 2007; Ashley, Grant & Grabov, 2005; Gierth, Mäser & Schroeder, 2005; Very et al., 2014). In plants, K+ transporters are classified into four major families: (1) HAK (high-affinity K+)/KUP (K+ uptake)/KT (K+ transporter), (2) Trk/HKT, (3) CHX (cation/hydrogen exchanger), and (4) efflux antiporters KEA (K+ efflux antiporter) (Gupta et al., 2008). Among them, the HAK/KUP/KT (HAK) family constitutes the largest K+ transporter family that is ubiquitously present in plant genomes with varied numbers, e.g., 13 genes present in Arabidopsis, 27 in maize, and 27 in rice (Ahn, Shin & Schachtman, 2004; Corratge-Faillie et al., 2010; Gupta et al., 2008; Rubio, Santa-Maria & Rodriguez-Navarro, 2000; Zhang et al., 2012).

HAK genes have been found to play key roles in plant development and stress-related responses. For instance, AtKUP4/TRH1 (Tiny Root Hairs 1) maintained the polar localization of AtPIN1 along with the auxin homeostasis and maxima in the root apex, thereby promoting root gravitropism response and root hair elongation (Rigas et al., 2001; Rigas et al., 2012; Vicente-Agullo et al., 2004). VvKUP2 (Vitis vinifera) promoted the expansion of berry epidermal cells (Davies et al., 2006). AtKUP2/SHY3 (Short Hypocotyl 3) mediated K+-dependent cell expansion in growing tissues, with the shy3-1 mutant plants having shorter hypocotyls, smaller leaves, and shorter flowering stems than the wild-type plants (Elumalai, Nagpal & Reed, 2002). Arabidopsis Kup2/6/8 triple mutants displayed larger plant bodies, thus suggesting their roles as negative regulators during the turgor pressure–dependent growth (Osakabe et al., 2013). Arabidopsis root meristem activity was maintained by KUP9 by regulating the K+ level and auxin homeostasis at a low K+ level (Zhang et al., 2020). HAK5 promoted the expression of INTEGRIN-LINKED KINASE1 (ILK1) to positively regulate plant innate immunity and abiotic stress response in Arabidopsis (Brauer et al., 2016). K+ deficiency stress induced the expression of rice OsHAK1, and when overexpressed in plants they display enhanced salt and drought tolerance (Chen et al., 2015; Chen et al., 2017; Chen et al., 2018). Additionally, constitutive overexpression or mutation analysis of OsHAK5, OsHAK21, and OsHAK16 demonstrated their role in K+ homeostasis and salt tolerance (Feng et al., 2019; Horie et al., 2011; Shen et al., 2015). HvHAK1 confers salt and drought tolerance in barley by enhancing the leaf mesophyll H+ homeostasis and improving K+ nutrition (Feng et al., 2020; Mangano, Silberstein & Santa-María, 2008).

In plants, the HAK family is the homolog of the bacterial K+ transporter KUP and fungal K+ transporter HAK (Bañuelos et al., 1995; Schleyer & Bakker, 1993; Very et al., 2014). Based on their hydropathy profiles, the plant HAK proteins were predicted to have 10–14 transmembrane (TM) domains, including a conserved K+ transport domain (GVVYGDLGTSPLY) ( Gierth & Mäser, 2007; Rodríguez-Navarro, 2000). Mutation assay analysis revealed that the role of K+ transport capacity is determined by both the 8th TM domain and the C-terminus of HAKs (Rodríguez-Navarro, 2000; Gomez-Porras et al., 2012; Mangano, Silberstein & Santa-María, 2008). Based on phylogenetic analysis, HAK family genes were generally classified into four clusters (I–IV) (Bañuelos et al., 1995; Gupta et al., 2008). HAK family members exhibit significant diversity in their subcellular localizations, including the plasma membrane, tonoplast, endoplasmic reticulum, and other endomembranes (Osakabe et al., 2013; Rigas et al., 2012). Expression analysis revealed that many members of the HAK family were also expressed in the root hairs and root tip cells, thereby implying the HAK family members are involved in K+ uptake (Ahn, Shin & Schachtman, 2004; Elumalai, Nagpal & Reed, 2002; Qin, Wu & Wang, 2019; Yang et al., 2014). Indeed, several HAK family members have been shown to participate in K+ uptake and translocation in a few model plants, including Arabidopsis, rice, barley, maize, and tomato (Very et al., 2014).

M. truncatula has been regarded as a model system for studying legume genetics and its relatively small genome size helps understand nodule symbiosis (Young et al., 2011). Despite the functional importance of the HAK genes, surprisingly little is known about their family members in M. truncatula. In this study, we performed comprehensive genome-wide analyses of the M. truncatula HAK family genes including phylogenetic relationships, chromosomal distributions, gene duplications, gene structures, cis-acting regulatory elements, and expression patterns in response to both K+ deficiency and abiotic stress. Finally, these results not only elucidated the structures and expression patterns of 20 MtHAKs genes but also laid the foundation for their future functional analysis in M. truncatula.

Materials and Methods

Identification and sequence analysis of MtHAKs

MtHAKs sequences were obtained from the Medicago truncatula genome databases (HAPMAP, https://medicagohapmap2.org). The amino acid (aa) sequences of Arabidopsis (TAIR, http://www.arabidopsis.org/) and rice (TIGR, http://rice.plantbiology.msu.edu/) HAKs were used as the reference sequences for searching predicted homolog sequences in M. truncatula using the HMMER3.0 software (http://hmmer.org/). Subsequently, the genes were screened using a threshold of <1e−100 E-value (full sequence and best one domain). Candidate protein members were verified using the SMART database (http://smart.embl-heidelberg.de/) and NCBI-Conserved Domain Database (CDD, https://www.ncbi.nlm.nih.gov/Structure/cdd/wrpsb.cgi) (Zhao et al., 2021), with proteins with shorter aa length (<400 aa) and those containing incomplete K+ transporter domains being discarded. The longest gene was chosen for further analysis only if it had alternative splicing variants. Subcellular localization of MtHAK proteins was predicted using the WOLF PSORT software (https://www.genscript.com/wolf-psort.html) and the TMHMM Server 2.0 online tool (https://services.healthtech.dtu.dk/service.php?TMHMM-2.0) was used for predicting the protein transmembrane helices.

Construction of MtHAKs phylogenetic tree

HAK protein sequences of Arabidopsis and rice were retrieved from the NCBI database (https://www.ncbi.nlm.nih.gov) (Table S1), while multiple sequence alignment was conducted using the ClustalW program (Version 2.1; http://www.clustal.org/). MEGA7.0 was used to construct the phylogenetic tree using the neighbor-joining method along with the bootstrap replicates being up to 1,000 (Liu et al., 2019; Liu et al., 2020).

Gene structure and conserved motif analysis

Gene structure and conserved motifs were visualized using the TBtools (Toolkit for Biologists integrating various biological data-handling tools) software (Chen et al., 2020). The conserved and identified motifs of protein sequences were predicted via the MEME (Multiple Expectation Maximization for motif Elicitation) program (Version 5.1.1), with the maximum protein motif number being set as 10, and the other parameters set as default (http://meme-suite.org/tools/meme) (Bailey et al., 2009).

Chromosomal location and synteny analysis

The MtHAK chromosomal location was illustrated by the circos diagram by annotating genes to their specific chromosomal location in their genome sequences by using the TBtools software. These syntenic analyses were carried out by using the MCScanX with gene duplication parameters (Wang et al., 2012).

Analysis of cis-acting regulatory elements in MtHAKs promoter regions

Putative cis-acting regulatory elements were analyzed using the PlantCARE online software (http://bioinformatics.psb.ugent.be/webtools/plantcare/html/). The 2.0 kb promoter sequences located upstream of the transcription starting site in each MtHAK gene were extracted from the M. truncatula genome database.

Analysis of microarray expression profile

The microarray data of the expression profiles of MtHAKs in the roots, vegetative bud, stem, petiole, leaf, flower, pods, and seeds and their responses to abiotic stress were obtained from the MtGEA (Benedito et al., 2008). When a gene corresponded to multiple probes, the maximum value of the probe was selected for the subsequent analysis. The normalized microarray data was used to create the heatmap through the TBtools software, based on the mean value of each gene expression in all the analyzed organs. The expression patterns of MtHAKs in response to salt, drought, and cold stresses were obtained from the NCBI under GEO accession number GSE136739 (Song et al., 2017). The expression abundance of each MtHAK gene was represented by fragments per kilobase million (FPKM). The relative stress-induced expression levels were calculated by comparing with the control samples. The clustered heatmap was generated using the TBtools software and based on their relative expression.

Stress treatment and qRT-PCR

For K+ deficiency stress treatment, two-week-old seedlings were grown in 1/2 Hoagland nutrient medium without K+ for 0 (control), 1, 6, 12, 24, and 48 h, respectively. For the salt stress treatment, two-week-old seedlings were grown in 1/2 Hoagland nutrient medium containing 300 mM NaCl for 0, 1, 6, 12, 24, and 48 h, respectively. For drought stress treatment, two-week-old seedlings were grown in 1/2 Hoagland nutrient medium containing 18% PEG6000 for 0, 1, 6, 12, 24, and 48 h, respectively. The root samples were subsequently cut, then snap frozen in liquid nitrogen, and finally stored at −80 °C until further use. The qRT-PCR analysis was performed in triplicates for each of the biological replicates. Their relative expression levels were calculated using the 2−ΔΔCt analysis method (Liu et al., 2019; Zhao et al., 2022). The expression levels of the control samples were normalized to one, with the MtActin gene being used as the internal control. Standard deviations and the significant differences were indicated by error bars and an asterisk (*) (p < 0:05), respectively.

Results

Identification of HAK members in M. truncatula

To identify M. truncatula HAK genes, we conducted a genome-wide search using the HMMER3.0 Software (http://hmmer.org/) based on the M. truncatula genome sequences along with the Arabidopsis and rice HAK genes as subjected queries. Then, we identified 20 nucleotide sequences with a typical canonical K+ transporter domain (Pfam accession no. PF02705) using the Pfam and SMART databases, and they were subsequently designated as MtHAK1 to MtHAK20 depending on their chromosomal positions (Table 1). Detailed information on the 20 HAK genes is listed in Table S1. The number of protein transmembrane segments (TMS) ranged between 10 and 13, with the most common being 12–13 (70%). All the examined HAK proteins were predicted to be mainly localized in the plasma membrane using a PSORT analysis (http://www.psort.org). The protein length of the 20 identified HAK proteins ranged from 619 aa (MtHAK3) to 856 aa (MtHAK2) with an average length of 778 aa. Their relative molecular weights (MW) varied from 69.03 kDa (MtHAK3) to 95.67 kDa (MtHAK2). The isoelectric points (pI) ranged from 5.44 (MtHAK8) to 9.39 (MtHAK19).

Table 1 Characteristics of MtHAK genes in M. truncatula.

Gene name	Gene ID	No. of
aa	MW (kDa)	pI	TMS	Subcellular
localization	
MtHAK1	Medtr2g008820.1	849	94.74	5.66	11	PM	
MtHAK2	Medtr2g438150.1	856	95.67	8.26	13	PM	
MtHAK3	Medtr2g438160.1	619	69.03	9.28	13	PM	
MtHAK4	Medtr3g094090.1	794	89.3	7.24	13	PM	
MtHAK5	Medtr4g094660.1	787	88	8.08	13	PM	
MtHAK6	Medtr4g099260.1	815	90.92	8.74	11	PM	
MtHAK7	Medtr5g034500.1	782	87.25	8.24	13	PM	
MtHAK8	Medtr5g070670.1	849	95.05	5.44	12	PM	
MtHAK9	Medtr5g071630.1	725	81.46	6.63	12	PM	
MtHAK10	Medtr5g071827.1	666	74.26	7.22	10	PM	
MtHAK11	Medtr5g071860.1	754	84.24	7.02	11	PM	
MtHAK12	Medtr6g007697.1	776	87.1	7.71	13	PM	
MtHAK13	Medtr6g033165.1	819	91.56	8.71	12	PM	
MtHAK14	Medtr7g108480.1	773	87.24	7.77	12	PM	
MtHAK15	Medtr8g022130.1	766	85.45	7.77	12	PM	
MtHAK16	Medtr8g063840.1	840	93.11	6.51	12	PM	
MtHAK17	Medtr8g063900.1	745	83.49	8.45	11	PM	
MtHAK18	Medtr8g088200.1	782	87.17	9.03	10	PM	
MtHAK19	Medtr8g099090.1	792	88.52	9.39	12	PM	
MtHAK20	Medtr8g107510.1	782	86.83	8.32	12	PM	
Note:

aa, amino acid; MW, molecular weight; pI, isoelectric points; TMS, transmembrane segments; PM, plasma membrane.

HAKs phylogenetic relationship among M. truncatula, Arabidopsis and rice

To analyze the evolutionary relationships of the MtHAK proteins, we conducted phylogenetic analyses of 60 HAK amino acid sequences (20 – M. truncatula, 13 – Arabidopsis, and 27 – rice) to construct a phylogenetic tree using the neighbor-joining method. According to the evolutionary tree, we classified all HAK members into four major groups: Groups I–IV. Furthermore, we classified the MtHAK proteins into three clusters (from I to III): Cluster I (MtHAK6, 14, 15, 17, and 18), Cluster II (MtHAK4, 7, 9, 10, 11, 12, 13, 19, and 20), and Cluster III (MtHAK1, 2, 3, 17, 5, and 8) (Fig. 1). All members in group IV belong to rice. The most members existed in Cluster II in M. truncatula, thus comprising 45% of all MtHAKs. The phylogenetic tree showed that MtHAKs were most closely related to Arabidopsis KUPs than those of rice HAKs, thereby indicating that MtHAKs might share evolutionary functional similarities with Arabidopsis KUPs. All MtHAKs in cluster I were distributed together with the already-identified AtHAK5, which suggested that they may be crucial for K+ uptake from a low-K+ level soil (Lara et al., 2020). Among cluster II members, MtHAK4 and MtHAK19 shared high sequence identity with AtKUP2 (Elumalai, Nagpal & Reed, 2002), and AtKUP4 (Rigas et al., 2001; Vicente-Agullo et al., 2004), respectively, thus implying they are likely to be involved in plant development processes. Additionally, among cluster III, MtHAK1 and MtHAK8 clustered together with AtKUP7 (Han et al., 2016), thereby suggesting their role in K+ acquisition and translocation under low K+ concentration.

Figure 1 Phylogenetic analysis of HAK proteins in M. truncatula (red circle), A. thaliana (green triangle), and O. sativa (blue square).

The tree was constructed using MEGA7.0 software by the neighbor-joining method. The numbers next to the branch represent the 1,000 bootstrap replicates expressed in percentage.

Gene structure and motif composition of MtHAK genes

MtHAK proteins were listed in order based on the phylogenetic analysis (Fig. 2A), which was consistent with the results in Fig. 1. Closely related members shared similar exon/intron structures, which were related to their biological functions. Gene structures of the MtHAKs were abtained based on the arrangement of the untranslated region, exon, and intron sequences generated using the TBTools software. As shown in Fig. 2B, the exon number of MtHAK genes varied from 8 to 10, and the longest exon existed in the end of a gene except for MtHAK2, which is consistent with previous reported data (He et al., 2012; Hyun et al., 2014). Additionally, most MtHAKs in the same cluster shared high exon-intron structure similarity (Fig. 2B).

Figure 2 Phylogenetic tree, gene structure, and conserved motifs of the HAKs in M. truncatula.

(A) Phylogenetic tree of the MtHAK proteins. (B) Exon-intron structure distribution. (C) Conserved protein motifs.

To study the structural features, we analyzed conserved protein motifs of MtHAKs using the MEME program. We identified the conserved protein motifs varying from 29 to 50 aa in lengths and designated them as motifs 1–10. Conserved protein motif information is shown in Table S2. The highly conserved K+ transport domain (GVVYGDLGTSPLY), included in motif 9, existed in all MtHAK proteins besides MtHAK10 (Fig. 2, Table S2). Motifs 1, 2, 3, 4, 5, 6, 7, 8, and 10 were almost evenly distributed along with a feature domain of K+ transporter (Fig. 2C, Table S2) in all the MtHAK proteins. Therefore, the motifs of conserved K+ transporter and similarities of gene structure in the same cluster together implied the closing function among these HAK members.

Chromosomal distribution and synteny analysis of MtHAK genes

All identified MtHAK genes were mapped onto chromosomes from the M.truncatula genome database to identify and locate their chromosomal distribution. Results showed that MtHAKs were distributed on seven of the eight chromosomes, with chromosome 8 containing the highest number of six MtHAK genes (Fig. 3). Five MtHAK genes were located on chromosome 5, three on chromosome 2, two on chromosome 4 and 6, one on chromosome 3 and 7, and no gene was allocated on chromosome 1 (Fig. 3). These results indicated that MtHAKs were scattered randomly onto different chromosome locations.

Figure 3 The synteny analysis of MtHAKs displayed between the M. truncatula and Arabidopsis genomes.

The M. truncatula and Arabidopsis chromosomes are represented by yellow and green boxes, respectively. Blue lines indicate the collinear relationship of MtHAKs between M. truncatula and Arabidopsis, while green lines indicate the MtHAK gene pairs.

We further performed synteny analysis between M. truncatula and Arabidopsis to verify the evolutionary relationships and history of the MtHAKs. Subsequently, we found seven collinear gene pairs between M. truncatula and Arabidopsis in the dataset (Fig. 3 and Table S3). This indicated that these identified genes might already have existed before protein structure divergence, thereby further implying a strong phylogenetic relationship. Furthermore, only one gene pair (MtHAK2/MtHAK5) existed as paralogs in M. truncatula.

Analysis of cis-acting elements in the promoter region of MtHAK genes

To further investigate the gene function and regulatory mechanism of MtHAKs, we analyzed the 2 kb regions upstream of the translation start site of the 20 MtHAK genes using the PlantCARE database. We identified 73 putative cis-elements in the MtHAK promoters based on functional annotation, and the major types of cis-elements are shown in Fig. 4 and Table S4. Post analysis, we identified cis-elements corresponding to different plant hormones like auxin (TGA-element and AuxRE-core), gibberellin (GARE-motif and P-box), MeJA (TGACG-motif and CGTCA-motif), ethylene (ERE-box), ABA (ABRE), and salicylic acid (TCA-element), in the promoter regions of all MtHAKs genes except MtHAK20, thereby suggesting that MtHAKs expression may be regulated by different phytohormones. Furthermore, we also found abiotic stress-responsive elements, including STRE, ARE, WRE3, WUN-motif, MBS, LTR, DRE-core, DRE1, and TC-rich repeats, in all the MtHAKs promoter regions except MtHAK20. Additionally, zein metabolism regulation element (O2-site), endosperm expression element (GCN4-motif and AACA-motif), palisade mesophyll cells element (HD-Zip 1), meristem expression element (CAT-box and CCGTCC-motif), and seed regulation element RY-element were also abundant in the promoter of MtHAKs except MtHAK20. However, the MtHAK20 promoter region was abundant in light-responsive elements (Table S4).

Figure 4 Analysis of the cis-acting regulatory elements in the promoter region of the MtHAK genes.

Depending on the functional annotation, the elements were classified into three main categories: phytohormone-responsive, abiotic stress-responsive, and plant growth and development-related. The frequency of these elements in the promoter region was represented by the numbers and the depth of the red color.

Spatial expression profiles of MtHAK genes

To gain further insights into the potential biological function of MtHAK genes, we used the publicly available microarray data of the Medicago truncatula Gene Expression Atlas (MtGEA, https://mtgea.noble.org/v3/) to investigate the temporal and spatial expression pattern of the MtHAKs. MtHAK4 showed relatively high expression in all tissues, while that of MtHAK18 was low in all tissue (Table S5). Notably, MtHAK6 and MtHAK16 were expressed preferentially in the roots, thereby implicating their role in K+ uptake from the soil (Fig. 5, Table S5).

Figure 5 Expression patterns of the MtHAK genes in different developmental tissues.

The microarray data were normalized based on the mean value of each gene in all the analyzed plant organs. The heat map was portrayed by the relative expressions after log2 transformed.

Both cluster III genes, MtHAK2 and MtHAK3 exhibited similar expression patterns and relatively high expression in leaves. MtHAK13 was exclusively and highly expressed in floral organs, whereas MtHAK8 showed the same in immature seeds (Fig. 5, Table S5). Interestingly, MtHAK5 and MtHAK12 exhibited high and gradually increased expression patterns during the reproductive stages and finally peaked at 24 days after pollination (DAP) Contrastingly, MtHAK15 was specifically highly expressed in immature seeds (10 DAP) with the expression pattern gradually decreasing along with seed maturation. Therefore, the spatial and temporal expression profiles indicated the functional diversity of MtHAK genes in Medicago trunculata development.

Expression patterns of MtHAK genes under K+ deficiency

Due to the major function of the HAK family being K+ transport, we investigated the expression profiles of MtHAK genes in the roots under K+ deficient conditions using qRT-PCR. As shown in Fig. 6, among the 20 MtHAK genes, we obtained eight genes that showed upregulated expression patterns post K+ deficiency treatment. MtHAK6, MtHAK7, and MtHAK17 expression slightly increased and finally peaked at 48 h post treatment. MtHAK15 and MtHAK18 showed nearly the same expression pattern at the five different time points. MtHAK9, MtHAK10, and MtHAK11 transcripts were strongly upregulated at 6 h, then peaking at 12 h and 24 h, and finally went down at 48 h. Therefore, these results suggested that these MtHAK genes were K+ deficiency-responsive. Furthermore, it was noteworthy that MtHAK6 was highly and specifically expressed in Medicago trunculata roots and also significantly upregulated in response to K+ deficiency.

Figure 6 Relative expression of the MtHAK genes in response to K+ deficiency treatment.

Two-week-old seedlings were placed in K+ deficient conditions for 0, 1, 6, 12, 24, and 48 h. Mean values and standard errors were calculated from three biological replicates. An asterisk (*) indicates the significant difference between K+ deficiency and control at p < 0.05.

Expression patterns of MtHAK genes under salt and drought stresses

Several HAK genes have been reported to participate in abiotic stresses (Elumalai, Nagpal & Reed, 2002; Vicente-Agullo et al., 2004; Chen et al., 2015; Shen et al., 2015). To verify this hypothesis, we evaluated the expression profiles of eight K+ deficiency responsive genes via qRT-PCR under salt and drought stress treatments. The results revealed that all eight genes were induced by salt and drought stresses to different extents (Figs. 7 and 8).

Figure 7 Relative expression of the MtHAK genes in response to salt stress.

Two-week-old seedlings were treated with 300 mM NaCl for 0, 1, 6, 12, 24, and 48 h. Mean values and standard errors were calculated from three biological replicates. An asterisk (*) indicates the significant difference between the salt-stressed and control at p < 0.05.

Figure 8 Relative expression of the MtHAK genes in response to drought stress.

Two-week-old seedlings were treated with 18% PEG6000 for 0, 1, 6, 12, 24, and 48 h. Mean values and standard errors were calculated from three biological replicates. An asterisk (*) indicates the significant difference between the drought-stressed and control at p < 0.05.

We determined the expression profile of MtHAK genes in Medicago trunculata roots at different times (0, 1, 6, 12, 24, and 48 h) under salt treatment (300 mM NaCl in nutrient solution). The results showed that expression of MtHAK7, MtHAK9, MtHAK15, MtHAK17, and MtHAK18 exhibited significant upregulation. Interestingly, MtHAK7 and MtHAK18 were quickly and continuously upregulated from 1 h and subsequently increased at 48 h (Fig. 7).

Additionally, we analyzed the expression profiles of MtHAK genes in Medicago trunculata roots under drought treatment simulated by 18% PEG6000 at different times (0, 1, 6, 12, 24, and 48 h). Under drought treatment, all selected genes besides MtHAK9 were upregulated, albeit to different levels at different times (Fig. 8). In particular, MtHAK10, MtHAK15, and MtHAK18 rapidly responded to dehydration at 1 h. Contrastingly, MtHAK17 was moderately upregulated from 6 to 48 h. Both MtHAK6 and MtHAK7 exhibited highly induced expression at 24 h.

Interestingly, we found that MtHAK15, MtHAK17, and MtHAK18 were strongly upregulated by both salt and drought stresses. The expression level of MtHAK18 increased rapidly at 1 h as compared to the control, under both salt and drought treatments (Figs. 7 and 8).

Discussion

HAK family genes play key roles not only in K+ acquisition and uptake, but also in plant growth, development, and abiotic stress response (Osakabe et al., 2013; Zhao et al., 2016). Although comprehensive genome-wide analysis of the HAK gene family has been widely reported in various plants, studies of the HAK gene family in the model legume M. truncatula were still lacking (Ahn, Shin & Schachtman, 2004; Gupta et al., 2008; Zhang et al., 2012). The release of the M. truncatula genome information makes it possible to systematically characterize and identify the HAK genes. In this study, we identified 20 HAK genes in M. truncatula. We characterized their genetic structures as well as their expression patterns in different tissues and also during stress responses.

We classified the 20 identified HAK members into three clusters (clusters I to III) based on the evolutionary relationships, which was consistent with the previous classification in Arabidopsis (Fig. 1) (Rubio, Santa-Maria & Rodriguez-Navarro, 2000). Phylogenetic analysis of HAK proteins revealed that MtHAKs shared higher similarity with AtHAKs as compared to the OsHAKs (Fig. 1), thereby suggesting that MtHAKs may share similar functionality with Arabidopsis AtHAKs. Gene structure analysis showed that MtHAK genes contained 8–10 exons, with the last exon being the longest. However, the exception was MtHAK2, which was consistent with the previously reported exon-intron structure of HAKs (He et al., 2012; Hyun et al., 2014). Conserved protein motif analysis indicated that all the identified MtHAKs had at least five typical K+ transporter motifs.

The tissue-specific gene expression patterns reflect their function and potential biological roles in plants. Approximately 10 of the 13 Arabidopsis AtHAK genes were strongly expressed in the root (Ahn, Shin & Schachtman, 2004). MtHAK6 was preferentially highly expressed in the roots, and it belonged to the same clades of AtHAK5 in the phylogenetic tree, which was also expressed in roots and mediated high-affinity root K+ uptake (Lara et al., 2020), thereby implicating their role in K+ acquisition from the soil (Fig. 5, Table S5). MtHAK16 shared high similarity with AtKUP12, which showed root hair-specific expression (Ahn, Shin & Schachtman, 2004). Therefore, these results may help elucidate the biological function of Arabidopsis orthologous MtHAK genes in K+ acquisition in M. truncatula.

Some plant HAK genes were shown to participate in plant growth and development. For instance, AtKUP4/TRH1 mutation impaired the root gravitropism response and root hair elongation (Rigas et al., 2001; Rigas et al., 2012; Vicente-Agullo et al., 2004). Knockout of AtKT2/KUP2 caused shorter hypocotyl length, small rosette leaves, and short flowering stem phenotype (Elumalai, Nagpal & Reed, 2002). MtHAK13 was exclusively and highly expressed in the floral organs, while showing low expression levels in other tissues, thereby suggesting its critical role in floral development. MtHAK5, MtHAK8, MtHAK12, and MtHAK15 were specifically and highly expressed during the reproductive stages, thus implying their roles in facilitating seed maturation and maintaining fertility. The varied tissue expression pattern of the MtHAK genes indicated their diverse functions in plants.

Under K+ deficiency conditions, plants maintain cytosolic K+ homeostasis by uptaking K+ through HAKs, and these K+ transporter genes represent a major transcriptional regulation mechanism during low- K+ stress. AtHAK5 and AtKT1 are two essential transporters mediating high-affinity K+ uptake in the Arabidopsis roots, with the roots of their double-mutant unable to sustain plant growth (Lara et al., 2020). ZmHAK5 was characterized as a high-affinity K+ transporter in maize (Qin, Wu & Wang, 2019). The expression of OsHAK1 and OsHAK5 were significantly upregulated in roots under low K+ conditions, thereby maintaining the K+ uptake and translocation from the root to the shoot (Chen et al., 2015; Chen et al., 2017; Chen et al., 2018; Yang et al., 2014). We found that K+ deficiency upregulated the root-specific expression of eight MtHAKs, especially MtHAK10 and MtHAK11 (Fig. 6). MtHAK6 was preferentially and highly expressed in the roots, and it increased under K+ deficiency stress (Fig. 6). Therefore, we expect that several HAK genes could increase the K+ absorption capacity during K+ deficiency.

Previous studies reported that HAK genes were crucial for regulating water potential and turgor pressure during osmotic adjustment. These genes also positively regulated plant stress responses by regulating the balance of K+ influx/efflux balance, e.g., OsHAK1 expression increases in the rice roots post the K+-deficient condition and it positively regulated the salt and drought stress tolerance response (Chen et al., 2015; Chen et al., 2017; Chen et al., 2018). Consistent with the above results, in our study, e.g., MtHAK7, MtHAK9, MtHAK15, MtHAK17, and MtHAK18 exhibited significantly upregulated expression levels in M. trunculata roots under salt stress (Fig. 7). Interestingly, many cis-acting elements related to phytohormones, plant growth and development, and abiotic stress response, were extensively distributed in the promoter regions of the MtHAKs (Fig. 4). ABREs are drought-stress responsive elements (Sah, Reddy & Li, 2016). MtHAK genes contained ABRE elements in their promoters, which further implied that MtHAKs participate in drought responses (Figs. 4 and 6). Moreover, the qRT-PCR analysis showed that the expressions of most of the selected MtHAK genes were noticeably upregulated after drought stress. Notably, MtHAK6, which was preferentially highly expressed in the roots, was also significantly upregulated post drought stresses (Fig. 8). In particular, the expression of MtHAK15, MtHAK17, and MtHAK18 were strongly and specifically upregulated in M. truncatula roots under K+ deficiency, salt, and drought stress conditions, thus implying that these genes are potential candidates for high-affinity K+ uptake while also being essential in salt and drought tolerance.

Conclusions

Based on phylogenetic analysis, we identified and characterized 20 MtHAK protein sequences from M. truncatula which were grouped into three clusters. Furthermore, we analyzed the chromosome location, conserved protein motif, and gene structure of all the M. truncatula HAK genes. The cis-acting elements regulating plant growth and development, or those responsive to phytohormone and abiotic stress were abundant in the promoter regions of MtHAKs. Gene expression analysis assay revealed that MtHAKs exhibited diverse tissue-specific expression patterns in various tissues using the publicly available RNA-seq data. Additionally, eight upregulated genes showed varied expression patterns post the K+ deficiency treatment. The expression pattern analysis under K+ deficiency, drought, and salt stress suggested that these genes are candidates for high-affinity K+ uptake that are also crucial in drought and salt tolerance. Therefore, these results provide the first genetic description of the K+ transporter family in M. truncatula, while also laying the foundation for molecular breeding of stress-resistant legume crops in the future.

Supplemental Information

Supplemental Information 1 Raw data of q-pcr.

Click here for additional data file.

Supplemental Information 2 HAK amino acid sequences of Medicago truncatula, Arabidopsis thaliana and Oryza sativa.

Click here for additional data file.

Supplemental Information 3 Conserved amino acid motifs and annotation of MtHAKs.

Click here for additional data file.

Supplemental Information 4 Duplications of MtHAK genes between Medicago truncatula and Arabidopsis thaliana.

Click here for additional data file.

Supplemental Information 5 Annotation of cis-acting regulatory elements in the promoters of MtHAK genes.

Click here for additional data file.

Supplemental Information 6 Microarray data of MtHAK genes in different organs and developmental stages.

Click here for additional data file.

We would like to thank the reviewers for their comments and suggestion.

Additional Information and Declarations

Competing Interests

Author Contributions

Data Availability

The authors declare that they have no competing interests.

Yanxue Zhao conceived and designed the experiments, performed the experiments, analyzed the data, prepared figures and/or tables, authored or reviewed drafts of the article, and approved the final draft.

Lei Wang analyzed the data, prepared figures and/or tables, and approved the final draft.

Pengcheng Zhao performed the experiments, analyzed the data, prepared figures and/or tables, and approved the final draft.

Zhongjie Liu performed the experiments, analyzed the data, prepared figures and/or tables, and approved the final draft.

Siyi Guo analyzed the data, authored or reviewed drafts of the article, and approved the final draft.

Yang Li analyzed the data, authored or reviewed drafts of the article, and approved the final draft.

Hao Liu conceived and designed the experiments, analyzed the data, authored or reviewed drafts of the article, and approved the final draft.

The following information was supplied regarding data availability:

The raw measurements are available in the Supplemental Files.

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
