# Peer review of "Genome-wide identification, characterization and expression analysis of HAK genes and decoding their role in responding to potassium deficiency and abiotic stress in Medicago truncatula"

_PeerJ, doi:10.7717/peerj.14034_

## Round 0.1 · original submission · Major Revisions

Thank you for the science interest and submission to PeerJ journal. Unfortunately, the manuscript got very critical remarks, even suggestion to reject this work. Extensive English editing is highly recommended. The reviewers did very detailed and constructive comments for the updates. Please take all the reviewers’ comments into account. I think the text should be rewritten, restructured and presented as new paper after major revision. Encourage manuscript resubmission.

Reviewer 1 ·

Basic reporting

The language needs extensive modification, some parts are hard to understand.

Experimental design

Experimental design is fit.

Validity of the findings

objectives are not clear to add valuable knowledge.

Additional comments

The manuscript entitled “Genome-wide identification, characterization and expression pattern analysis of HAK gene family in Medicago truncatula under abiotic stresses” by Zhao et al, performed bioinformatic identification of HAK gene family under various abiotic stress, in silico. One of the major drawbacks is experimental design of this manuscript. Author used RNA-seq data for various stress and then selected few genes for K stress. What is the logic for choosing K stress? Why not performed same stresses as like RNA-seq.
The manuscript needs extensive modification before reaching conclusion to publish.
The abstract is randomly written please re-write it with flow.
The objective of study is not clearly stated.
Line 29-31, 123-125, is not clear. Please rewrite this sentence.
Line 115-117, what is protein with error? Please re-write this sentence
Line 118, how alternative splicing variant was selected?
Line129-132 how Gene structure and protein conserved motifs were analyzed by TBtools software? In the next line, Secondly, how the conserved and identified motifs identified with MEME different from those identified with TBTools?
Line 133 what is “Synteny Correlation Analysis”? I never heard about this. Please explain it.
Line 135-136, how syntenic relationship was identified and what is mean by “syntenic relationship verification”?
Line 146, link is not found please provide valid website.
Line 143-154, author used RNA seq data for tissue specific expression and expression under abiotic stresses. I suggest author should perform RT-qPCR for validation of expression data obtained.
Line 160-161 what is “untreatment expression levels”? I suggest authors to used standard terms.
For subcellular localization author stated that genes are localized in membrane, It not possible to predict membrane localization of protein using ordinary GFP. I suggest author should use membrane associate markers to validate their exact location.
The manuscript is not well structed and experimental design is not valid. I must reject this manuscript until all suggested modifications are incorporated …………………………………………………..

Reviewer 2 ·

Basic reporting

This manuscript describes the genome-wide analyses of the Medicago HAK gene family. HAK gene expression was further investigated using public data, and the expression some HAK members was validated by qRT-PCR. Although the English writing is understandable, it is suggested that it should be polished by English native speakers. Raw data was supplied and the research purpose is clear, but the method described should be consistent.

Experimental design

The method description (L108-111) is not consistent with the result section (L174-176). In the method section, the author stated that they used HMMER3.0 to search homologs, but in the results BLASTP was mentioned. Please describe the identification process clearly but concisely.

Validity of the findings

There is quite a lot of information regarding the gene family of HAK, functional and mechanism study. The literature analysis should not be limited to Arabidopsis and rice. For instance, there is similar work on soybean. The authors should include it into this work, as both Medicago truncatula and soybean belong to the same family. Again, if there is detailed functional or even mechanism study for a specific HAK gene that is homology to a member from Medicago truncatula, that would give clues to the function of the particular gene. There are even HAK members functionally studied in Medicago truncatula, which also should be included.
In the subcellular localization analysis of MtHAK proteins, a co- localization marker should be included.

Additional comments

Minor points:
Latin names of plant species, and gene names should be italic, particularly in the Reference section.
L77-78: Not consistent with the context.
L88-90: HAK subcellular localization was diverse, then why all the MtHAKs were predicted to localize in PM, is this result consistent with their orthologues in Arabidopsis and rice?
L117: K+ potassium—duplication.
L120: was used for the transmembrane structure?
L124-125: change to Multiple sequence alignment was conducted through ClustalW program
L125: to construct
L129: the source and version of gene model annotation should be provided.
L134: annotating?
L143: Expression Profiles Analysis of MtHAKs Based on Transcriptome Data
L150: from the NCBI under GEO accession number GSE136739
L151: first appearance of FPKM
L160: a reference is needed for 2-ΔΔ analysis method.
L232: to analysis?
L242: functional annotation?
L282: Therefore
L287-288: grammar errors
L294: expression
L324: investigated should be replaced by indicated/suggested…
L345: K+ homeostasis?
L351: delete "-"

---

## Round 0.2 · Minor Revisions

Thanks for the updates and detailed answers. The reviewers have no more critical comments. However, there are some typos in the text (probably due to extensive text editing).

I suggest checking the English again, preferably by a proficient speaker.
Correct minor typos (spaces between words). Please see

Line 24: “HAK (high-affinity K+)/KUP (K+uptake)/KT (K+transporter) (HAK) family...” -
It is too long phrase listing all the protein names. And ‘Hak’ is repeated twice. Please update it like
“HAK transporter protein family (also known as HAK (high-affinity K+)/KUP (K+uptake)/KT (K+transporter) ) is...” or show synonyms n the next sentence to avoid complex wording in the beginning of the phrase.

Line 27: ‘Medicago truncatula(M. truncatula)’ - not need repeat species name in shorter form here.
It is worthy add here and in the Introduction section a phrase why this species is important to study (model organism, agrobiology value )?

Line 28: “still poorly known” - it is not best text. Remove word ‘poorly’. Please change to “still not studied in detail” or like that

Line 29-32: “was conducted.
... were extensively explored
... were assayed” - please change passive voice in English to direct forms like:
“We conducted... we assayed...”
It is about authors’ research, need to show your work.

Line 31: “extensively explored” - change wording to ‘explored in detail” or “analysed”
‘Explored’ word is not appropriate here

Line 42: ‘provide potential genes..’ - change phrase to
‘provide information about’ or ‘suggest list of genes’. It is about information, not genes itself.

Lines 52-52: ‘HAK (high-52 affinity K+)/KUP (K+up-take)/KT (K+transporter), Trk/HKT, CHX (cation/hydrogen 53 exchanger), and efflux antiporters KEA (K+efflux antiporter) ...’ -
This wording is too complex/ Just write all the families in 1) 2) 3) 4) or change, at least, full name’ HAK (high-52 affinity K+)/KUP (K+up-take)/KT (K+transporter)’ to shorter form here.

Line 107: ‘(http://www.medicagohapmap.org/)’ -add here access date, the database name. Make more detailed links, where it is possible.

Line 112: ‘data base’ - fix as ‘database’

Lines 127-128: ‘TBtool’ ‘MEME’ - it is good to give abbreviations (program names) in full.

Line 134: ‘MCScanX’ - need a reference for this tool.

Line 150: ‘(0 h)’ - please update wording. 0 hours after what?
Please fix the phrase “which were grouped into three clusters base on phylogenetic analysis.”
Write it in new sentence. Change ‘base on’ to ‘based on’.
Line 386: ‘A MG, and B PM. 2007.’ - write full authors’ names in this reference.

Table 1:
‘CharacteristicsofMtHAKgenes’ - typo. Separate the words by spaces.

Figure 2 Legend:
“Phylogenetictree”

Figure 3 Legend:
MtHAKswas ... M. truncatulaand ...Arabidopsisgenomes.
.... Arabidopsischromosomes

Figure 4:
“elementsanalysis ...
... classifiedinto”

Figure 5:
“log2transformed” - add space or ‘-‘ sign

Reviewer 1 ·

Basic reporting

The manuscript has been well improved so I have no more comment

Experimental design

The manuscript has been well improved so I have no more comments

Validity of the findings

The manuscript has been well improved so I have no more comment.

---

## Round 0.3 · accepted · Accept

Thanks for the update. All the remarks were taken into account. I endorse this publication in its current form.